# Exact invariant solution reveals the origin of self-organized oblique turbulent-laminar stripes

Florian Reetz[1], Tobias Kreilos[1] & Tobias M. Schneider[1]

Wall-bounded shear flows transitioning to turbulence may self-organize into alternating turbulent and laminar regions forming a stripe pattern with non-trivial oblique orientation. Different experiments and flow simulations identify oblique stripe patterns as the preferred solution of the well-known Navier-Stokes equations, but the origin of stripes and their oblique orientation remains unexplained. In concluding his lectures, Feynman highlights the unexplained stripe pattern hidden in the solution space of the Navier-Stokes equations as an example demonstrating the need for improved theoretical tools to analyze the fluid flow equations. Here we exploit dynamical systems methods and demonstrate the existence of an exact equilibrium solution of the fully nonlinear 3D Navier-Stokes equations that resembles oblique stripe patterns in plane Couette flow. The stripe equilibrium emerges from the well-studied Nagata equilibrium and exists only for a limited range of pattern angles. This suggests a mechanism selecting the non-trivial oblique orientation angle of turbulent-laminar stripes.

---

[1] Emergent Complexity in Physical Systems Laboratory (ECPS), École Polytechnique Fédérale de Lausanne, CH 1015 Lausanne, Switzerland. Correspondence and requests for materials should be addressed to T.M.S. (email: tobias.schneider@epfl.ch)

The complex laminar-turbulent transition in wall-bounded shear flows is one of the least understood phenomena in fluid mechanics. In the simple geometry of plane Couette flow (PCF), the flow in a gap between two parallel plates moving in opposite directions, the transitional flow spontaneously breaks the translational symmetries in both the streamwise and the spanwise direction causing regions of turbulent and laminar flow to coexist in space[1–5]. Remarkably, the flow may further self-organize into a regular pattern of alternating turbulent and laminar stripes[6–12] also observed in Taylor-Couette[6, 13–17] and channel flow[18–22]. The wavelength of these stripes or bands is much larger than the gap size, the only characteristic scale of the system, and they are oblique with respect to the streamwise direction. Consequently, both the large-scale wavelength and the oblique orientation of turbulent-laminar stripes must directly follow from the flow dynamics captured by the governing Navier–Stokes equations. Experiments and numerical flow simulations reliably generate stripe patterns but a theory explaining the origin of the pattern characteristics is still missing. This is related to the Navier–Stokes equations being highly nonlinear partial differential equations, whose theoretical analysis remains challenging.

It was the early observation that an oblique turbulent-laminar pattern can be the preferred solution of the Navier-Stokes equations that motivated R. Feynman to stress the lack of "mathematical power [of his time] to analyze [the Navier–Stokes equations] except for very small Reynolds numbers"[23]. Recent advances in numerical methods not only allow the simulation of flows but also the construction of exact equilibria, traveling waves and periodic orbits of the fully nonlinear 3D Navier–Stokes equations. These exact invariant solutions are believed to be embedded in a strange invariant set generating the chaotic dynamics of turbulent flow in the system's state space[24]. Consequently, a picture emerges where turbulent flow is described as a chaotic walk between dynamically unstable invariant solutions which together with their entangled stable and unstable manifolds support the turbulent dynamics[25]. Exact invariant solutions are thus "building blocks" which resemble characteristic flow structures that are observed in flow simulations and experiments, when the dynamics transiently visits the exact invariant solution. A theoretical explanation of oblique stripe patterns within this dynamical systems description requires the as yet unsuccessful identification of exact invariant solutions resembling the detailed spatial structure of turbulent-laminar stripes, including their oblique orientation and large-scale periodicity.

Nagata discovered the first invariant solution of PCF[26–28]. Like most invariant solutions of PCF found since then[29], this so-called Nagata equilibrium is periodic in the streamwise and spanwise directions, repeating on the scale of the gap height. Such periodic solutions do not capture the coexistence of turbulent and laminar flow on scales much larger than the gap height and consequently cannot underly oblique stripes. Spanwise localized invariant solutions[30, 31] and doubly localized invariant solutions in extended periodic domains[32] show nonlinear flow structures coexisting with laminar flow but no known invariant solution captures oblique orientation or suggests a pattern wavelength matching oblique stripe patterns.

We present a fully nonlinear equilibrium solution of PCF (Fig. 1b), resembling the oblique stripe pattern observed in direct numerical simulations (Fig. 1a). Parametric continuation demonstrates that this stripe equilibrium is connected to the well-studied Nagata equilibrium via two successive symmetry-breaking bifurcations, and that its existence is limited to oblique orientations.

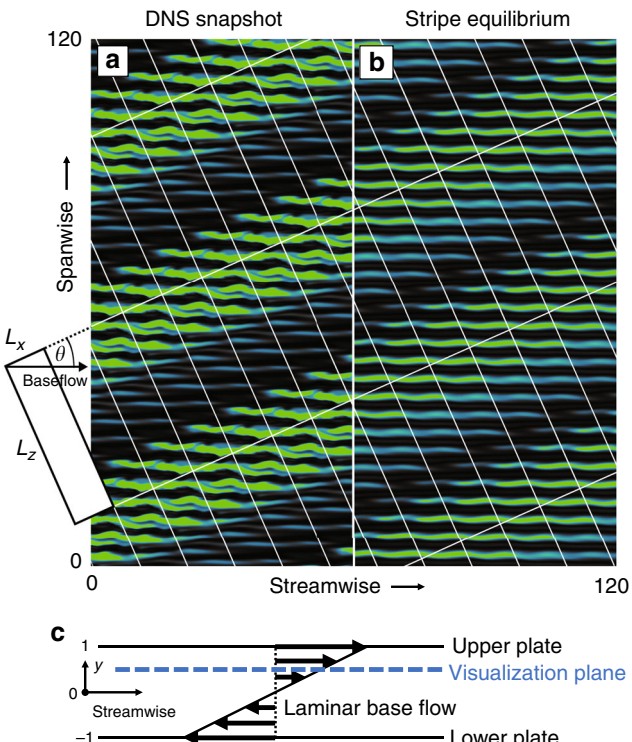

**Fig. 1** Equilibrium solution underlies turbulent-laminar stripes. **a** The self-organized pattern of oblique turbulent-laminar stripes is observed in direct numerical simulations of plane Couette flow at Re = 350. Following ref. [7] a tilted $x$-$z$-periodic domain outlined on the left side with $(\theta, L_x, L_z) = (24°, 10, 40)$ is used for computations. **b** The observed stripe pattern is captured by an exact invariant equilibrium solution of the fully nonlinear 3D Navier–Stokes equations. The contours are turbulent kinetic energy saturating at $\mathbf{u}^2 = 0.25$ (green), where $\mathbf{u}$ is the velocity fluctuation field around the laminar base flow (**c**). The plane of visualization is at 3/4 of the`` gap height

## Results

**Simulating stripe patterns.** For direct numerical simulations (DNS) of oblique stripe patterns in PCF we use a parallelized version of the pseudo-spectral code CHANNELFLOW[25, 33]. The numerical domain is periodic in two perpendicular dimensions along the plates ($x$ and $z$) with periods of $(L_x, L_z) = (10, 40)$ in units of half the gap height. No-slip boundary conditions are imposed at the moving plates located at $y = ±1$. Inversion symmetry with respect to the domain center is enforced. The relative plate velocity and the associated base flow are tilted against the periodic domain dimensions at an angle of $\theta = 24°$ following Barkley and Tuckerman[7]. At Reynolds number $Re = Uh/v = 350$, with the relative plate velocity $2U$, gap height $2h$ and kinematic viscosity $v$ the flow organizes into self-sustained turbulent-laminar stripes, as shown in Fig. 1a, where we periodically repeat the computational domain to highlight the large-scale structure of the pattern.

**Equilibrium resembling stripes.** An invariant equilibrium solution capturing the stripes was found by introducing a large-scale amplitude modulation to a known spatially periodic equilibrium using a suitable window function, similar to ref. [31]. Specifically, the Nagata equilibrium was periodically extended in the spanwise direction for $n = 9$ periods, then sheared to align the velocity streaks with the base flow in the tilted domain and finally multiplied with a scalar window function equal to a scaled mean field

of turbulent kinetic energy of the oblique stripe pattern from several DNS runs. Using the constructed velocity field as initial guess, Newton iteration yields the stripe equilibrium (Fig. 1b).

The stripe equilibrium shares the small-scale wavy modulation with the Nagata equilibrium but also shows the large-scale oblique amplitude modulation of the turbulent-laminar stripe pattern. The amplitude modulation between the high-amplitude turbulent region and the low-amplitude laminar region of the equilibrium on average follows a sinusoidal profile closely resembling the pattern mean flow found in DNS at identical boundary conditions[8]. The stripe equilibrium moreover captures detailed features of the turbulent-laminar interfaces. A base flow directed into a turbulent region leads to a sharper "upstream" interface than a base flow directed out of a turbulent region at a "downstream" interface. The direction of the base flow is reversed for $y \rightarrow -y$. An upstream interface in the upper half thus corresponds to a downstream interface in the lower one. This gives rise to so-called overhang regions[1, 34] and an asymmetry between the left and right interface in Fig. 1 where turbulent kinetic energy is visualized at $y = 0.5$ above the midplane. Finally, the stripe equilibrium is symmetric under inversion $\sigma_i[u, v, w](x, y, z) = [-u, -v, -w] (-x, -y, -z)$, a symmetry also found for the mean flow of stripe patterns[8]. The sinusoidal amplitude modulation, the captured overhang regions and the inversion symmetry, all characteristic of the pattern's mean flow, together with the visual comparison in Fig. 1 show that the stripe equilibrium has the spatial features of the oblique stripe pattern. We have thus identified a first exact invariant solution underlying oblique turbulent-laminar patterns.

The unstable eigenspace of the evolution operator linearized around the equilibrium is spanned by 15 directions. The remaining $\sim 10^6$ directions are attracting. Consequently, the dynamics is attracted towards the stripe equilibrium from almost all directions. The exponential growth rates $\omega_r$ along the unstable directions are small compared to typical turbulent time scales in the flow, given by the oscillatory frequencies $\omega_i$ in the spectrum of eigenvalues (Fig. 2). The low dimensional unstable eigenspace and the small exponential growth rates suggest a weakly unstable exact invariant solution that is a dynamically relevant transiently visited "building block" of the chaotic saddle underlying the turbulent flow.

**Origin of the equilibrium.** At small scales, the stripe equilibrium reflects the wavy streak structure of the spatially periodic Nagata equilibrium. This suggests that the stripe equilibrium emerges from the Nagata equilibrium in a bifurcation creating oblique

long-wavelength modulations. To identify this pattern-forming bifurcation numerically, the Nagata equilibrium needs to "fit" in an extended tilted periodic domain aligned with the wave-vector of the neutral mode creating the oblique long-wavelength modulation. The Nagata equilibrium indeed not only satisfies the streamwise and spanwise periodic boundary conditions of the commonly studied minimal flow domain but may also be periodic with respect to selected larger tilted domains. The symmetry group of the Nagata equilibrium, including all combined discrete translations over streamwise-spanwise periods ($\lambda_{st}, \lambda_{sp}$), intersects with the group of translations of a tilted rectangular domain, with periodicity ($L_x, L_z$), if

$$L_x = \frac{k\lambda_{st}}{\cos\theta} = \frac{l\lambda_{sp}}{\sin\theta}, \quad L_z = \frac{m\lambda_{st}}{\sin\theta} = \frac{n\lambda_{sp}}{\cos\theta} \quad (1)$$

is satisfied for $(k, l, m, n) \in \mathbb{N}$ and $0° < \theta < 90°$. Geometrically, condition (1) describes how the $x$–$z$ coordinate lines of the tilted domain wind on a torus defined by the streamwise-spanwise periodic minimal domain. The condition is satisfied if the coordinate lines are closed curves (Fig. 3a). For the domain ($\theta, L_x, L_z$) = (24°, 10, 40) considered so far, the geometric condition (1) implies wavelengths ($\lambda_{st}, \lambda_{sp}$) = (1.02, 4.06) at which the Nagata equilibrium does not exist. Keeping $L_z = 40$ and choosing winding

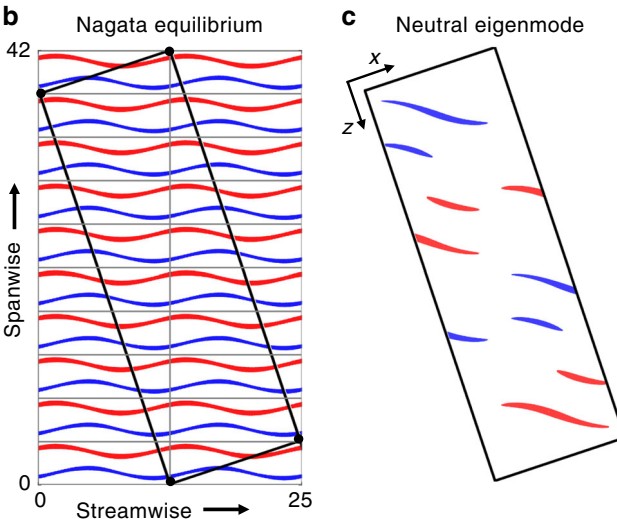

**a** Coiled coordinate lines

**b** Nagata equilibrium    **c** Neutral eigenmode

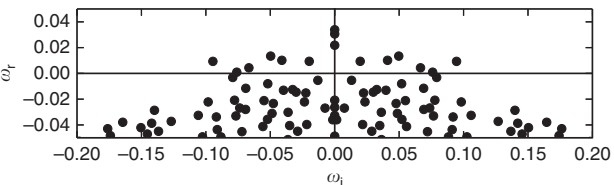

**Fig. 2** Eigenvalue spectrum characterizing the linear stability of the stripe equilibrium. Shown are the 100 leading eigenvalues in the complex plane ($\omega_r, \omega_i$), computed by Arnoldi iteration at Re = 350[33]. 15 eigenvalues with a positive real part $\omega_r > 0$ quantify the exponential growth rate $\omega_r$ along the unstable directions in the linear eigenspace of the stripe equilibrium. The imaginary parts $\omega_i$ quantify the oscillatory frequencies in the eigenspace and the absolute values indicate typical turbulent time scales in the flow in units of $U/h$. The wide aspect ratio of the spectrum (uniform axes) graphically illustrates that the maximum growth rate $\omega_r = 0.034$ is small compared to the typical turbulent time scales. The stripe equilibrium shown in Fig. 1b is thus weakly unstable

**Fig. 3** Instability of the Nagata equilibrium creates oblique amplitude modulations. **a** Torus representing a streamwise–spanwise periodic domain. If tilted rectangular coordinate lines $x$ and $z$ (black) close on themselves, all solutions on the torus also respect the periodicity of a domain spanned by those lines. Specifically, the Nagata equilibrium with streamwise–spanwise periodicity ($\lambda_{st}, \lambda_{sp}$) = (12.65, 4.22) (gray lines) is also periodic with respect to the tilted domain (black) with ($\theta, L_x, L_z$) = (18.4°, 40/3, 40), shown in **b** for $Re_l = 164$. In this tilted domain, a bifurcation with neutral eigenmode (**c**) can be detected at $Re_l$. This bifurcation introduces oblique long-wavelength amplitude modulations on the Nagata equilibrium. Red (blue) contours represent positive (negative) streamwise velocity in the midplane

numbers $(k, l, m, n) = (1, 1, 1, 9)$ however leads to a slightly modified domain $(\theta, L_x, L_z) = (18.4°, 40/3, 40)$ in which the Nagata equilibrium with $(\lambda_{st}, \lambda_{sp}) = (12.65, 4.22)$ exists, as displayed in Fig. 3b. On the lower branch of the Nagata equilibrium close to the saddle-node bifurcation, there is a pitchfork bifurcation at $Re_I = 164$. Its neutrally stable long-wavelength eigenmode, whose eigenvalue changes sign at $Re_I$, is plotted in Fig. 3c. This is the initial pattern-forming bifurcation creating oblique amplitude modulations on the Nagata equilibrium.

Using parametric continuation we follow both the periodic Nagata equilibrium (named $\mathcal{A}$ hereafter) and the emerging modulated equilibrium solution ($\mathcal{B}$) from its primary bifurcation point at $(Re, \theta, L_x)_I = (164, 18.4°, 40/3)$ to the parameters $(Re, \theta, L_x)_C = (350, 24°, 10)$ of the stripe equilibrium ($\mathcal{C}$). In the three dimensional parameter space we choose a continuation path parametrized by tilt angle $\theta$ with the Reynolds number linear in $\theta$, such that $Re(\theta) = (Re_I(\theta_C - \theta) + Re_C(\theta - \theta_I))/(\theta_C - \theta_I)$ and domain length $L_x(\theta) = L_z/(n \tan(\theta))$ for $n = 9$ and constant domain width of $L_z = 40$. The resulting bifurcation diagram demonstrates that the Nagata equilibrium $\mathcal{A}$, is connected to the stripe equilibrium $\mathcal{C}$ (Fig. 4).

The primary bifurcation is of pitchfork type, subcritical, forward in $Re$ and breaks the streamwise–spanwise translation symmetry of $\mathcal{A}$. Along the bifurcating branch of $\mathcal{B}$ significant amplitude modulations of the small-scale periodic signal form with period $L_z/2$ along $z$, as indicated by the double-pulse profile of the $z$-dependent and $x$–$y$-averaged fluctuations $|u|(z)$ at $Re = 225$ in Fig. 4b. The modulation period reflects a discrete translation symmetry $\sigma_B$ over half the domain diagonal,

$\sigma_B[u, v, w](x, y, z) = [u, v, w](x + L_x/2, y, z + L_z/2)$. Equilibrium $\mathcal{B}$ inherits this symmetry from $\mathcal{A}$ because $\sigma_B$ is not broken by the neutral mode of the primary bifurcation (Fig. 3c).

A secondary pattern-forming bifurcation occurs at $(Re, \theta, L_x)_{II} = (332, 23.4°, 10.3)$ along solution branch $\mathcal{B}$ (blue line in Fig. 4). This subcritical pitchfork bifurcation breaks the translation symmetry $\sigma_B$. The spatial period of the amplitude modulation is doubled and gives rise to solution branch $\mathcal{C}$ forming a single-pulse equilibrium. Solution branch $\mathcal{C}$ (red line in Fig. 4) reaches $Re_C = 350$ after undergoing an additional saddle-node bifurcation at $(Re, \theta, L_x) = (243, 20.8°, 11.7)$. The amplitude profiles of single-pulse and double-pulse equilibria show that the single-pulse with period $L_z = 40$ has large modulations at $Re = 350$ and $\theta = 24°$, while the modulations in the double-pulse equilibrium are reduced (Fig. 4c). This agrees with the observations that stripes tend to have pattern wavelengths $\lambda$ in the range of $40 \leq \lambda \leq 60$ at $Re$ around $350$[6, 10]. In summary, two bifurcations successively break discrete translation symmetries of the Nagata equilibrium to create the stripe equilibrium solution.

Small-scale velocity streaks carry a wavy modulation which has a streamwise phase that is clearly evident when plotting the streamwise vorticity at the midplane. We illustrate streamwise wave fronts by lines connecting vorticity maxima or minima in the spanwise direction (red/blue lines in Fig. 5). Straight and

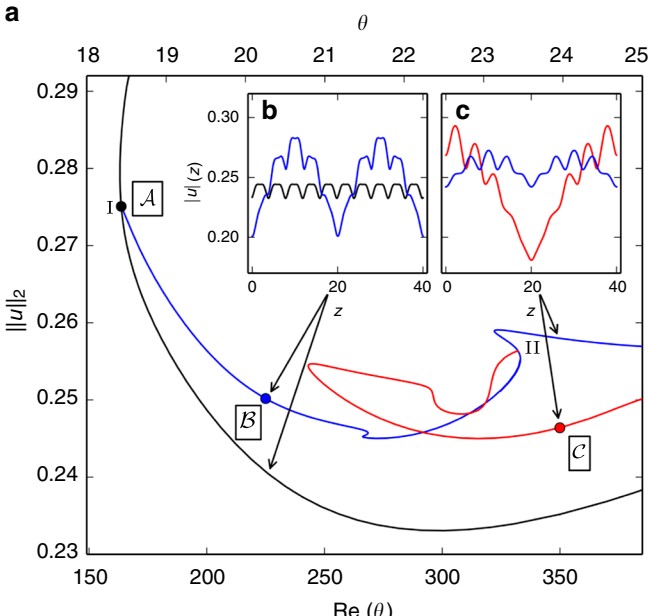

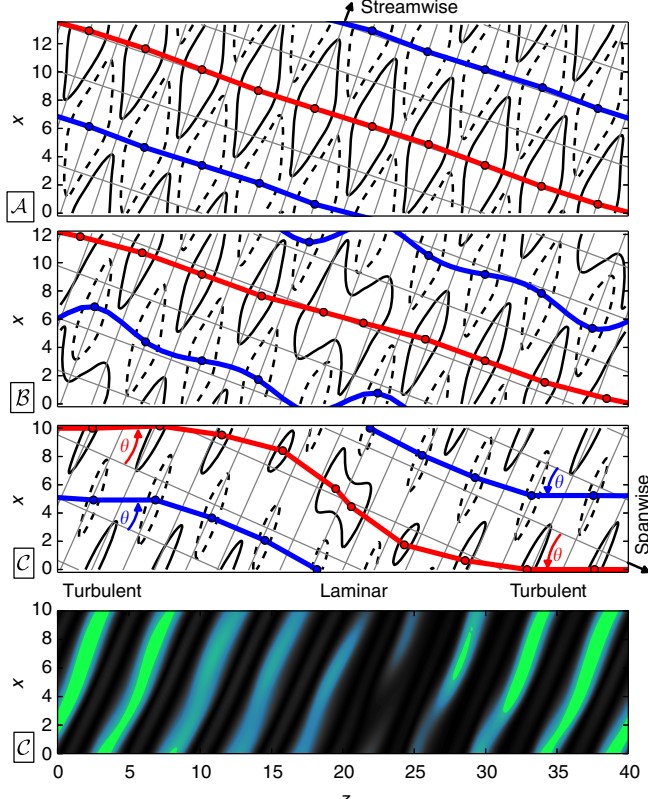

**Fig. 4** Pattern forming bifurcations give rise to the stripe equilibrium. **a** A sequence of pattern-forming bifurcations from the small-scale periodic Nagata equilibrium $\mathcal{A}$ (Fig. 3b) leads to the large-scale modulated stripe equilibrium $\mathcal{C}$ (Fig. 1b). The solution branches are plotted in terms of the domain averaged velocity square $||u||_2 = (2L_xL_z)^{-1/2}(\int \mathbf{u}^2 dxdydz)^{1/2}$ over linearly coupled bifurcation parameters $\theta$ (top axis) and $Re(\theta)$ (see text). A primary pattern-forming bifurcation on $\mathcal{A}$ at $(Re, \theta, L_x)_I = (164, 18.4°, 40/3)$ creates equilibrium $\mathcal{B}$ with double-pulse profile of $x$-$y$ averaged squared velocity $|u|(z) = (2L_x)^{-1/2}(\int \mathbf{u}^2 dxdy)^{1/2}$ (inset panel **b**). A secondary pattern-forming bifurcation at $(Re, \theta, L_x)_{II} = (332, 23.4°, 10.3)$ creates the single-pulse solution branch of equilibrium $\mathcal{C}$ (inset panel **c**). Points mark the exact invariant solutions shown in Fig. 5

**Fig. 5** Phase modulations along the bifurcation sequence. Streamwise vorticity $\omega_{st} = (\cos(\theta)\mathbf{e}_x + \sin(\theta)\mathbf{e}_z) \cdot \nabla \times \mathbf{u}$ at the midplane (black solid/dashed contours at $\omega_{st} = \pm 0.12$) encodes streamwise phase information of wavy streak modulations for equilibria $\mathcal{A}$, $\mathcal{B}$ and $\mathcal{C}$ along the bifurcation sequence (points in Fig. 4). Red (blue) lines connecting vorticity maxima (minima) represent wave fronts of constant streamwise phase. The Nagata equilibrium $\mathcal{A}$ has wave fronts oriented in the spanwise direction. The stripe equilibrium $\mathcal{C}$ has sigmoidal wave fronts. In the turbulent region the wave fronts are oriented at $\theta = 24°$ (red/blue arrows), and align with the pattern wave vector (in the $z$-direction). Bottom panel indicates turbulent and laminar regions in $\mathcal{C}$ (see also Fig. 1b)

strictly spanwise oriented wave fronts indicate a constant streamwise phase of all streaks of the (spanwise periodic) Nagata equilibrium $\mathcal{A}$. The primary pattern forming bifurcation from $\mathcal{A}$ to $\mathcal{B}$ introduces local phase shifts which dislocate vorticity extrema away from a straight alignment and bend the wave front. The dislocations introduced into $\mathcal{B}$ are symmetric with respect to half-domain translations $\sigma_{\mathcal{B}}$ and centered at $z = 0$ and $z = 20$. For the stripe equilibrium $\mathcal{C}$ formed from $\mathcal{B}$ in the second pattern forming bifurcation, the topology of the wave fronts is preserved but they are geometrically deformed into sigmoidal structures.

In the turbulent region of equilibrium $\mathcal{C}$, the wave fronts of the wavy streaks are skewed at $\theta = 24°$ against the spanwise direction and oriented exactly along the pattern wave vector ($\mathcal{C}$ in Fig. 5). Assuming that all exact invariant solutions underlying stripe patterns show this alignment of wave fronts in the turbulent region with the pattern orientation, we conjecture that the range of possible skewing angles Nagata-type equilibria can sustain[35] limits the range of angles at which oblique stripe patterns can exist.

**Pattern angle selection**. We identified equilibrium $\mathcal{C}$ at pattern angle $\theta = 24°$. Continuation in $\theta$ for fixed pattern wavelength $\lambda = 40$ determines the range in $\theta$ for which the stripe equilibrium exists. At $Re = 350$, the equilibrium exists in the range $16.5° \leq \theta \leq 26.1°$ before it undergoes saddle-node bifurcations (Fig. 6). Outside this range the stripe equilibrium is not sustained. The range of orientation angles over which the stripe equilibrium exists agrees with the range over which oblique stripe patterns of wavelength $\lambda = 40$ are observed in simulations[8]. At lower $Re$, the range of allowed pattern angles shrinks and shifts towards larger values (Fig. 6). This trend of allowed $\theta$ for varying $Re$ aligns with

simulations and experiments of turbulent-laminar stripes[8]. The finite existence range of the fully nonlinear exact equilibirum solution of the 3D Navier–Stokes equations thus appears to select the non-trivial angle at which self-organized turbulent-laminar stripes emerge in transitional shear flows.

## Discussion

Experimental and numerical observations of self-organized oblique turbulent-laminar stripes in wall-bounded extended shear flows suggest the existence of exact invariant solutions underlying these patterns. We present the first such invariant solution of the fully nonlinear 3D Navier–Stokes equations in plane Couette flow that captures the detailed spatial structure of oblique stripe patterns. The stripe equilibrium emerges from the known Nagata equilibrium via a sequence of two pattern-forming bifurcations with long-wavelength oblique neutral modes. The existence of the stripe equilibrium at wavelength $\lambda = 40$ is limited to oblique orientations in a finite range of pattern angles around $\theta = 24°$. The existence range agrees with simulations and experimental observations of turbulent-laminar stripes. This suggests a selection mechanism for the pattern angle and provides a route towards explaining why turbulent-laminar stripes are oblique.

## Data availability

The fully resolved velocity field of the stripe equilibrium is provided as a data file in the supplementary material. The file format is NetCDF and can be directly imported in many post-processing and visualization tools. Below, we specify the names of grid variables in the file and how to reproduce the equilibrium using the open source software CHANNELFLOW. The velocity field is numerically resolved in a domain of size $(L_x, H, L_z) = (10, 2, 40)$ using a grid of size $(N_x, N_y, N_z) = (42, 33, 340)$. The variables of the grid dimensions are named "X", "Y" and "Z", where "Y" is the wall-normal dimension. The components of the velocity vectors along each dimension are called "Velocity_X", "Velocity_Y", and "Velocity_Z". The equilibrium solution can be checked for convergence and further analyzed using CHANNELFLOW. The Reynolds number of $Re = 350$ for plane Couette flow must be set using the flag "-R 350". To define the tilt angle of $\theta = 24°$ of the laminar base flow with respect to the domain dimensions, the flag "-theta 0.4182" must be specified. Numerical stability requires a CFL-number in the range of $0.2 < CFL < 0.4$. The provided data file allows to reconstruct the data supporting the findings of this study. The data is also available from the corresponding author upon reasonable request.

## Code availability

All numerical algorithms used in the present study are open source and can be downloaded via https://www.channelflow.ch.

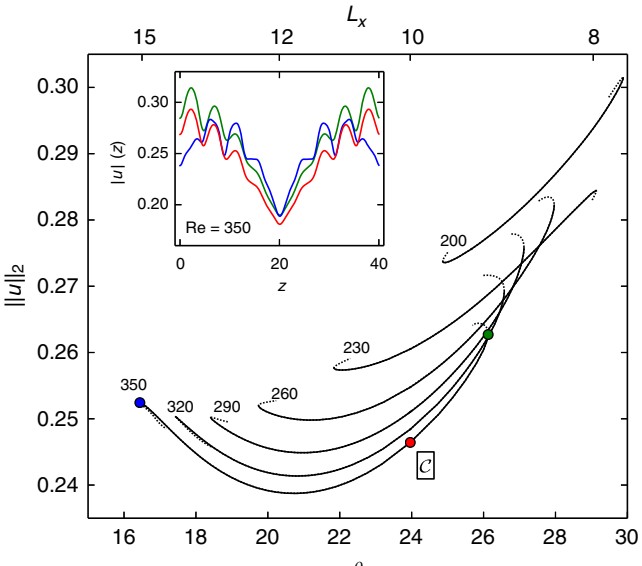

**Fig. 6** Range of pattern orientation angle $\theta$ for which the stripe equilibrium $\mathcal{C}$ exists. Parametric continuations in $\theta$ at constant $L_z = 40$ implying $L_x \sim \tan^{-1}(\theta)$ (top axis). The solution branches are labeled by $Re$, which remains fixed for each continuation. Beyond saddle-node bifurcations only the initial part of the branches are plotted (dotted lines). Along the remaining parts of the branches (not shown), the equilibrium solutions no longer represent the stripe pattern. The inset displays the amplitude profiles at selected points along the branch (like in Fig. 4). For $\theta$ increasing towards the upper saddle-node the amplitude of the profile $|u|(z)$ rises globally at all $z$; for $\theta$ decreasing towards the lower saddle-node the amplitude maximum decreases. At intermediate angles close to $\theta = 24°$ the equilibrium best represents turbulent-laminar stripes

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

## Acknowledgements

This work was supported by the Swiss National Science Foundation (SNF) under grant no. 200021-160088.

## Author contributions

T.M.S. conceived the project. F.R. carried out the research under supervision of T.M.S. and with initial input of T.K. The manuscript was prepared by F.R. and T.M.S., in correspondence with T.K.

## Additional information

**Competing interests:** The authors declare no competing interests.

