## [peer review file · Nature Communications]

Reviewer #1 (Remarks to the Author):

The authors reported an invariant solution that may correspond to the non-trivial oblique stripe pattern of localized turbulence in the subcritical transition process of wall-bounded shear flows, which was found by tracking well-known equilibrium solution. Their tracking of an invariant equilibrium solution was initiated from the spatially-periodic steady solution of plane Couette flow, the so-called Nagata's solution, and performed with using tilted and one-dimensionally-elongated domains. During their tracking, the path of equilibrium solution undergoes two bifurcations, through which there arise large-scale modulations (in the longer direction of the domain). Actually, the authors successfully demonstrated a single-pulse solution branch that emerges after the secondary pattern forming bifurcation, exhibits a reasonable wave length and vector (i.e., the angle of obliqueness) similar to those of the turbulent stripe in the plane Couette flow obtained by DNS (direct numerical simulation w/o any turbulence model nor symmetric conditions). The turbulent stripe itself was found in the early 2000s, but its pattern-formation mechanism including the obliqueness and its selection has been unclear. Currently, we can find many literature that reported the turbulent stripe in various canonical flows and high-Re flows under stable stratification, and always the stripe pattern exhibited very large-scale modulations compared to the gap size and inherent turbulent eddies, as shown also in the present paper. However, we had still lacked a theoretical explanation of the existence of the turbulent stripe. The present novel finding by the authors may provide a great impact on the longstanding physics problem, the subcritical transition of turbulence, like the Nagata's solution, and shed light on some of mysteries about the pattern-formation mechanism.

The present manuscript is convincing and understandable, and their solution tracking as well as numerical simulations seems to have been done with cares. Basically, I'd like to recommend its publication, but let me suggest some issues to be addressed before the acceptance.

- Regarding the title: I feel "How turbulence self-organizes into oblique ..." would be misleading. The present paper/finding provides "a solution" of the fully nonlinear 3D Navier-Stokes equations, which just "resembles" the turbulent stripe, at a single value of Reynolds number. A possibility of the existence of the turbulent stripe is proven now, but the self-organization process/mechanism from the dynamics aspect is not discussed in the present manuscript. Alternatives are "A solution of oblique ..." or "An invariant solution of oblique ...".
- An additional text or figure of plane-Couette configuration should be included: for instance, Barkley & Tuckerman (2005, PRL) said "plates located at $y \pm h$ move at velocities $\pm U \langle x \rangle / b$ ". Otherwise, some readers cannot figure out the situations and some sentences: for instance, "a plane at 3/4 of the gap height", "The direction of the base flow is reversed for y ", "symmetric under inversion", and " u_{rms} ".
- Please provide Re for Fig. 2b and 2c. How did the authors detected the state given in Fig. 2c.
- In the present study, the wavelength of large-scale (largest?) modulated solution is fixed at 40. This value might be suitable for Re = 350 in PCF, but the pattern wavelength should increase with the decreasing Reynolds number, as you may know or the existing studies reported. (Equilibria B and C in Fig.3 seem inconsistent with this tendency, but I'm afraid that such an intuitive insight might be wrong. Anyway, the present study fails to discuss the Reynolds number tendencies of the orientation angle and wavelength.) How do the authors think about the Reynolds-number dependency of the present solution?
- Definitions of "x-y averaged u_{rms} " in Fig. 3 caption is unclear. Is "u" the fluctuating component or not? Is the " u_{rms} " on the vertical axis of Fig. 3 averaged also in time?
- Make the green arrows and fonts (θ) in Fig. 4 highlighted more for easy identification.

Reviewer #2 (Remarks to the Author):

Report on
How turbulence self-organizes into oblique turbulent-laminar stripes
by Florian Reetz, Tobias Kreilos & Tobias M. Schneider

Patterns of well-ordered, long-wavelength oblique turbulent-laminar bands are among the more exotic and mysterious phenomena observed wall-bounded shear flows. Reetz et al demonstrate what is almost surely the dynamical-systems origin for these patterns, via two pitchfork and one saddle-node bifurcation. I recommend that this article be published after the authors have addressed one major and several minor remarks.

Major remark: Reetz et al show that these patterns exist between angles of approximately 17° and 30° degrees and state that this agrees with [8]. However, [8] shows ranges in which these patterns are *observed experimentally* and *numerically in large non-tilted domains*, meaning that they (or turbulent version of them) are stable in large domains. In contrast, references [7] and [8] show such patterns in minimal tilted domains of the kind used in the present manuscript for angles ranging from 15° to 70° . This demonstrates the *existence* of such patterns over a much larger parameter range, whether or not they are stable in a large realistic domain. This much larger parameter range of existence would seem to be the relevant one for comparisons with the current manuscript, not the smaller range. The authors should address this discrepancy.

My minor comments are as follows. Many are optional.

“non-trivial oblique orientation” → “oblique orientation”

“the Nagata equilibrium”: Waleffe and Busse and Clever are also often cited in this context

“stripes, must” → “stripes must”

“allow to simulate flows” → “allow the simulation of flows”

“These dynamically unstable exact invariant solutions lead to a description of turbulence as a chaotic walk among invariant solutions.”

I wouldn't quite state the theory in this way. The theory from which the chaotic walk is predicted is not a consequence of the existence of many unstable solutions. A system can have many unstable solutions but not execute a chaotic walk between them. In fact, the Navier-Stokes equations always have many unstable solutions in most circumstances above certain Reynolds numbers, even when the solution observed is merely steady or periodic.

“the yet unsuccessful” → “the as yet unsuccessful”

“Nagata equilibrium is periodic in the streamwise and spanwise directions”

So are the patterns in Figures 1, 2 and 4, aren't they? Perhaps the authors mean that they are periodic with a fairly small period in these directions?

“domains, show” → “domains show”

“guess Newton” → “guess, Newton”

“spanned by 15 directions only”

Although much less than 10^6 , 15 unstable directions still sounds like a lot!

To me, a neutral eigenmode is one whose corresponding eigenvalue is zero over a range of Re because it corresponds to motion along one of the symmetry directions. After considering Figure 2 and the fact that the bifurcations leading to the final pattern are pitchforks and saddle-nodes, I realized that the eigenmode in Figure 2 is what I would call a bifurcating eigenmode (for which

the eigenvalue crosses zero transversely), not a neutral one.

“suggests the stripe equilibrium to emerge” → “suggests that the stripe equilibrium emerges”

“periodic at (L_x, L_z) ” → “with periodicity (L_x, L_z) ”

“subcritical, forward”

These terms are not always used in the same way by all researchers. The authors should explain that they (presumably) mean that

–the new branches (\mathcal{B}) are created for $Re > Re_I$ (forward)

–and that the parent branch (\mathcal{A}) is more stable for $Re > Re_I$ (subcritical),

i.e. for $Re > Re_I$, where branch \mathcal{B} exists, it is less stable than branch \mathcal{A} .

But if this is so, why should branch \mathcal{C} (arising from \mathcal{B}) be more observable than \mathcal{A} (the pure Nagata equilibrium, never spontaneously observed in simulations or experiment)? So perhaps the authors mean something else by subcritical?

“The amplitude profiles of single- and double-pulse equilibria show that the single-pulse with period $L_z = 40$ has large modulations . . . at $Re = 350$ and $\theta = 24^\circ$. . .

Solution branch \mathcal{C} (red line in Fig. 3) reaches $Re_C = 350$ after undergoing an additional saddle-node bifurcation at $(Re, \theta, L_x) = (243, 20.8^\circ, 11.7)$.”

For clarity, I would re-order to:

“Solution branch \mathcal{C} (red line in Fig. 3) reaches $Re_C = 350$ after undergoing an additional saddle-node bifurcation at $(Re, \theta, L_x) = (243, 20.8^\circ, 11.7)$.”

The amplitude profiles of single- and double-pulse equilibria

at $Re = 350$ and $\theta = 24^\circ$

show that the single-pulse with period $L_z = 40$ has large modulations . . .”

“is not broken by the neutral mode of the primary bifurcation”

Again, rather than “the neutral mode” I would say “by the bifurcating mode” or simply “by the primary bifurcation”.

“in spanwise direction” → “in the spanwise direction”

“alignement” → “alignment”

“allows to determine” → “determines”

Reviewer #3 (Remarks to the Author):

This paper reports the subharmonic instability and the resulting bifurcation on the Nagata (1990) steady solution to the Navier-Stokes equation as well as the subsequent second bifurcation on the found solution branch for plane Couette flow. This second bifurcation leads to a novel large-scale-patterned steady solution to the Navier-Stokes equation, which well describes the spatial property of oblique turbulent bands observed in transitional plane Couette flow. The key contribution of this paper is theoretical identification of the origin of a large-scale oblique pattern appearing in subcritical transition to turbulence in plane Couette flow. It should be highly appreciated because this kind of theoretical identification had not been achieved for commonly observed large-scale turbulence patterns in experiments or numerical simulations. Therefore I could recommend this paper for publication in Nature Communications if the following two comments are taken into account for the revision of the paper:

1. The instability of the novel solution in figure 1 (right) and figure 4 (bottom)

Although in page 3 (left), the authors briefly mentioned the instability of this solution, the detailed information should be necessary in the paper. As mentioned there are 15 unstable directions in phase space, but no information has not been provided on the eigenvalues which are indispensable for identification of significance of the instability of the found solution. How large are they with respect to the typical time scale? Without this information we cannot say that it is "weakly unstable". In addition, the corresponding eigenstructure to the leading (or more) eigenvalue(s) would be mentioned in order to discuss the relevance of the found solution with turbulence dynamics (turbulence dynamics could be considered to be an escaping process out of the invariant coherent solution).

2. Figure 5 and corresponding statement

This figure is important to interpret the selection mechanism of the inclination angle of oblique bands. The authors have shown the turning points suggesting the smaller and larger angle ends; however, there might be possibilities that the solution branches would further turn in the decreasing (increasing) direction on the smaller (larger) end. Have the authors confirmed that the solution branches are closed? If so, this should be mentioned. If not, the statement would be weaker than in the present form.

Response to Reviewer #1

The authors reported an invariant solution that may correspond to the non-trivial oblique stripe pattern of localized turbulence in the subcritical transition process of wall-bounded shear flows, which was found by tracking well-known equilibrium solution. Their tracking of an invariant equilibrium solution was initiated from the spatially-periodic steady solution of plane Couette flow, the so-called Nagata's solution, and performed with using tilted and one-dimensionally-elongated domains. During their tracking, the path of equilibrium solution undergoes two bifurcations, through which there arise large-scale modulations (in the longer direction of the domain). Actually, the authors successfully demonstrated a single-pulse solution branch that emerges after the secondary pattern forming bifurcation, exhibits a reasonable wave length and vector (i.e., the angle of obliqueness) similar to those of the turbulent stripe in the plane Couette flow obtained by DNS (direct numerical simulation w/o any turbulence model nor symmetric conditions). The turbulent stripe itself was found in the early 2000s, but its pattern-formation mechanism including the obliqueness and its selection has been unclear. Currently, we can find many literature that reported the turbulent stripe in various canonical flows and high-Re flows under stable stratification, and always the stripe pattern exhibited very large-scale modulations compared to the gap size and inherent turbulent eddies, as shown also in the present paper. However, we had still lacked a theoretical explanation of the existence of the turbulent stripe. The present novel finding by the authors may provide a great impact on the longstanding physics problem, the subcritical transition of turbulence, like the Nagata's solution, and shed light on some of mysteries about the pattern-formation mechanism.

The present manuscript is convincing and understandable, and their solution tracking as well as numerical simulations seems to have been done with cares. Basically, I'd like to recommend its publication, but let me suggest some issues to be addressed before the acceptance.

We thank reviewer #1 for this encouraging feedback and the suggestions.

1. Remark.

Regarding the title: I feel "How turbulence self-organizes into oblique ..." would be misleading. The present paper/finding provides "a solution" of the fully nonlinear 3D Navier-Stokes equations, which just "resembles" the turbulent stripe, at a single value of Reynolds number. A possibility of the existence of the turbulent stripe is proven now, but the self-organization process/mechanism from the dynamics aspect is not discussed in the present manuscript. Alternatives are "A solution of oblique ..." or "An invariant solution of oblique ...".

Response.

The title was chosen to highlight the importance of the discussed invariant solution for the formation of the stripe pattern. The pattern-forming instabilities discussed in the bifurcation scenario and the selection of the pattern orientation angle from a finite range of angles are two key mechanisms in the self-organization of turbulent-laminar stripes. We however agree with reviewer #1 that the title may suggest that we describe the temporal evolution of the self-organization process from an unpatterned state to the oblique stripe pattern.

To avoid potential confusion with a description based on such a temporal evolution, we have reconsidered the title following the suggestions of reviewer #1 and changed

the title in the revised manuscript to “Exact invariant solution reveals the origin of self-organized oblique turbulent-laminar stripes”. We understand that this title more explicitly refers to the main result in this manuscript: the existence of a first invariant solution resembling oblique stripes patterns and the exact bifurcation scenario from an unpatterned state to the oblique stripe pattern.

2. *Remark.*

An additional text or figure of plane-Couette configuration should be included: for instance, Barkley & Tuckerman (2005, PRL) said “plates located at $y = \pm h$ move at velocities $\pm Ux$ ”. Otherwise, some readers cannot figure out the situations and some sentences: for instance, “a plane at $3/4$ of the gap height”, “The direction of the base flow is reversed for y ”, “symmetric under inversion”, and “ u_{rms} ”.

Response.

We agree with reviewer #1 that introducing the system configuration of plane Couette flow (PCF) in more detail makes the manuscript easier accessible to the general reader. Following the suggestion of reviewer #1, we included the following descriptive sentence in the section “simulating stripe patterns”:

“No-slip boundary conditions are imposed at the moving plates located at $y = \pm 1$.”

Moreover, we added a sketch of the PCF base flow configuration in Fig. 1. This sketch also outlines the “plane at $3/4$ of the gap height”.

3. *Remark.*

Please provide Re for Fig. 2b and 2c. How did the authors detected the state given in Fig. 2c.

Response.

The Reynolds number was added to the caption of Fig. 3 (former Fig. 2). The state shown in Fig. 3c is found from a numerical stability analysis of equilibrium \mathcal{A} . Technically, we used Arnoldi iteration to compute this neutral eigenmode. A reference to Arnoldi iteration is added to the caption of the new Figure 2, where we present the eigenvalue spectrum in response to reviewer #3.

4. *Remark.*

In the present study, the wavelength of large-scale (largest?) modulated solution is fixed at 40. This value might be suitable for $Re = 350$ in PCF, but the pattern wavelength should increase with the decreasing Reynolds number, as you may know or the existing studies reported. (Equilibria \mathcal{B} and \mathcal{C} in Fig. 3 seem inconsistent with this tendency, but I'm afraid that such an intuitive insight might be wrong. Anyway, the present study fails to discuss the Reynolds number tendencies of the orientation angle and wavelength.) How do the authors think about the Reynolds-number dependency of the present solution?

Response.

We fix the period of stripe equilibrium \mathcal{C} at $L_z = 40$ because this period matches the pattern wavelength observed in simulations and experiments at $Re = 350$. The remark of reviewer #1 refers to equilibrium \mathcal{B} with a period of $L_z = 20$. This state does not resemble turbulent-laminar stripes and must not be compared to the experimental and numerical observations referenced by reviewer #1 because the pattern never forms at wavelengths of $\lambda = 20$. Equilibrium \mathcal{B} is an intermediate state in the bifurcation scenario that connects the unpatterned state \mathcal{A} with the stripe equilibrium \mathcal{C} via period doubling in space.

The question of detailed wavelength selection in stripe patterns, as raised by reviewer #1, requires a comparison between stripe equilibrium solutions at different wavelengths, Reynolds numbers and angles. The relevant solutions for stripes at larger wavelength (expected at smaller Re) will not be smoothly connected to the solution discussed here. We expect solutions with a different number of internal streaks to underlie stripes at larger wavelengths. It is not possible to simply ‘stretch’ the present stripe equilibrium to larger wavelengths because the internal structure of the stripe equilibrium, i.e. the wavy velocity streaks, do not allow this. Therefore, we need to leave it to future research to construct additional equilibrium solutions that resemble oblique stripe patterns at larger wavelengths.

We disagree with reviewer #1 that “the present study fails to discuss the Reynolds number tendencies of the orientation angle”. This discussion is included in section “pattern angle selection” and is based on Fig. 6 (former Fig. 5). There we say: “At lower Re , the range of allowed pattern angles shrinks and shifts towards larger values (Fig. 6). This trend of allowed θ for varying Re aligns with simulations and experiments of turbulent-laminar stripes⁸.”

5. *Remark.*

Definitions of “x-y averaged u_{rms} ” in Fig. 3 caption is unclear. Is “u” the fluctuating component or not? Is the “ u_{rms} ” on the vertical axis of Fig. 3 averaged also in time?

Response.

Since we show time-independent flows in Fig. 4 (previously 3) and Fig. 6 (previously 5), root-mean-square velocity u_{rms} refers to space averaged squared velocity. To avoid confusion with temporally fluctuating quantities, we changed the notation in the revised manuscript to a normalized L_2 -norm $\|u\|_2$ which we define explicitly for both, the global average and the z -dependent average, in the caption of Fig. 4.

About the question “Is u the fluctuating component or not?”. Yes, the vector field \mathbf{u} refers to the velocity fluctuations around the laminar base flow. We added a sentence at the first instance where \mathbf{u} appears, which is in the caption of Fig. 1:

“..., where \mathbf{u} is the velocity fluctuation field around the laminar base flow.”

6. *Remark.*

Make the green arrows and fonts (θ) in Fig. 4 highlighted more for easy identification.

Response.

We thank reviewer #1 for this suggestion and have changed the color of the labels in Fig. 5 (previously 4).

Response to Reviewer #2

Patterns of well-ordered, long-wavelength oblique turbulent-laminar bands are among the more exotic and mysterious phenomena observed wall-bounded shear flows. Reetz et al. demonstrate what is almost surely the dynamical-systems origin for these patterns, via two pitchfork and one saddle-node bifurcation. I recommend that this article be published after the authors have addressed one major and several minor remarks.

We thank reviewer #2 for the appreciation and the thorough remarks and suggestions.

1. *Major remark.*

Reetz et al show that these patterns exist between angles of approximately 17° and 30° degrees and state that this agrees with [8]. However, [8] shows ranges in which these patterns are observed experimentally and numerically in large non-tilted domains, meaning that they (or turbulent version of them) are stable in large domains. In contrast, references [7] and [8] show such patterns in minimal tilted domains of the kind used in the present manuscript for angles ranging from 15° to 70° . This demonstrates the existence of such patterns over a much larger parameter range, whether or not they are stable in a large realistic domain. This much larger parameter range of existence would seem to be the relevant one for comparisons with the current manuscript, not the smaller range. The authors should address this discrepancy.

Response.

Reviewer #2 raises the question on how the present results compare to Ref. [7] and [8]. The range of pattern angles reported in the cited work is not directly comparable to the one in the present work. The numerical setup in the cited work, a tilted domain of size $(L_x, L_z) = (10, 120)$, allows to simulate turbulent-laminar stripes of pattern wavelengths $\lambda = 40$, $\lambda = 60$ and $\lambda = 120$. Reviewer #2 is right that these authors find larger pattern angles of $\theta > 30^\circ$, but only for patterns of wavelengths $\lambda = 60$ or $\lambda = 120$. We explicitly choose to consider a pattern wavelength of $\lambda = 40$ and therefore choose a domain of size $(L_x, L_z) = (10, 40)$ in this study. Consequently, the range of pattern angles presented in Section “pattern angle selection” must not be compared with the larger range of pattern angles in Ref. [7] and [8] obtained in a domain three times larger than the one considered here.

A meaningful comparison between the simulated stripe pattern in Ref. [7] and [8] and the equilibrium stripe pattern discussed here, needs to be restricted to patterns of wavelength $\lambda = 40$, the period of the stripe equilibrium. The existence over different pattern angles of the simulated pattern with wavelength $\lambda = 40$ can be found in Fig. 29 in Ref. 8. This figure summarizes numerical and experimental results on the existence of the oblique stripe pattern and indicates the pattern wavelength. Patterns of wavelength $\lambda = 40$ cover the range $12.5^\circ \leq \theta < 30^\circ$ (cross markers and dark shaded region), which agrees well with the range of existence of the stripe equilibrium in the submitted manuscript.

The influence of numerical domains restricting patterns to a specific set of wavelength (wavelengths λ equal or smaller than the domain period L_z), is an important technical detail that – based on the reviewer’s comments – we wish to highlight when comparing the results on the range of pattern angles. Therefore, we added a footnote. Here we explicitly explain the different numerical setups and how to compare the present with the cited results.

In section “pattern angle selection”:

“The range of orientation angles over which the stripe equilibrium exists agrees with the range over which oblique stripe patterns of wavelength $\lambda = 40$ are observed in simulations³⁶.”

In Reference 36 (footnote):

“Figure 29 in reference 8 shows the range of parameters (Re, θ) over which oblique turbulent-laminar stripes are simulated in a tilted domain with periodicity $L_z = 120$. The authors distinguish three different pattern wavelengths. Patterns of wavelength $\lambda = 40$ are found in the interval $12.5^\circ \leq \theta < 30^\circ$.”

2. *Remark.*

“non-trivial oblique orientation” → *“oblique orientation”*

Response.

We highlight that the question of what selects the pattern orientation is not given by any obvious or trivial mechanism like boundary conditions or the direction of a body force. Instead the stripe pattern chooses a “non-trivial oblique orientation”. In our humble opinion, the attribute “non-trivial” is appropriate to describe the oblique orientation of the stripe pattern which neither aligns with a symmetry or any other *a priori* selected direction in the system.

3. *Remark.*

“the Nagata equilibrium”: *Waleffe and Busse and Clever are also often cited in this context.*

Response.

We have added the references to the work of Clever & Busse (1992) and Waleffe (1998) to the first instance of appearance in the manuscript (in the introduction). These two studies have independently found this invariant solution at a later time. For brevity, we continue with referring to this equilibrium by its common name “Nagata equilibrium” (as common in the field, see e.g. Ref. 29).

4. *Remark.*

“stripes, must” → *“stripes must”*

Response. Changed according to the suggestion of reviewer #2.

5. *Remark.*

“allow to simulate flows” → *“allow the simulation of flows”*

Response. Changed according to the suggestion of reviewer #2.

6. *Remark.*

“These dynamically unstable exact invariant solutions lead to a description of turbulence as a chaotic walk among invariant solutions.”

I wouldn't quite state the theory in this way. The theory from which the chaotic walk is predicted is not a consequence of the existence of many unstable solutions. A system can have many unstable solutions but not execute a chaotic walk between them. In fact, the Navier-Stokes equations always have many unstable solutions in most circumstances above certain Reynolds numbers, even when the solution observed is merely steady or periodic.

Response.

We thank reviewer #2 for pointing this out and we agree with the remark that the causality of this sentence is not sufficiently clear. We have changed the sentence to “Based on these dynamically unstable *exact invariant solutions* one may describe turbulence as a chaotic walk among invariant solutions...”

7. *Remark.*

“the yet unsuccessful” → “the as yet unsuccessful”

Response. Changed according to the suggestion of reviewer #2.

8. *Remark.*

“Nagata equilibrium is periodic in the streamwise and spanwise directions”

So are the patterns in Figures 1, 2 and 4, aren't they? Perhaps the authors mean that they are periodic with a fairly small period in these directions?

Response.

Reviewer #2 is right that we compare invariant solutions that are periodic in the streamwise and the spanwise direction over small scales in contrast to flows with turbulent-laminar patterns over large spatial scales. The latter also may be periodic. To eliminate any ambiguity, we modified the sentence according to the suggestions of reviewer #2:

“... this so-called Nagata equilibrium is periodic in the streamwise and spanwise directions, repeating on the scale of the gap height.”

9. *Remark.*

“domains, show” → “domains show”

Response. Changed according to the suggestion of reviewer #2.

10. *Remark.*

“guess Newton” → “guess, Newton”

Response. Changed according to the suggestion of reviewer #2.

11. *Remark.*

“spanned by 15 directions only”

Although much less than 10^6 , 15 unstable directions still sounds like a lot!

Response.

The purpose of this paragraph is to explain why the stripe equilibrium is of dynamical relevance. With 15 unstable eigendirection compared to $\sim 10^6$ stable eigendirections, the stripe equilibrium attracts the dynamics from almost all directions. Besides the number of unstable eigendirections also their linear growth rates define the stability of the stripe equilibrium. We discuss growth rates in detail in response to remark 1 of reviewer #3.

We understand that 15 unstable directions “sounds like a lot” compared to edge states with a single unstable direction. Since the assessment of 15 dimensions as “only” might depend on the relative perspective of the reader and it is not necessary for the purpose of this paragraph, we decided to remove the word “only”.

12. *Remark.*

To me, a neutral eigenmode is one whose corresponding eigenvalue is zero over a range

of Re because it corresponds to motion along one of the symmetry directions. After considering Figure 2 and the fact that the bifurcations leading to the final pattern are pitchforks and saddle-nodes, I realized that the eigenmode in Figure 2 is what I would call a bifurcating eigenmode (for which the eigenvalue crosses zero transversely), not a neutral one.

Response.

In linear stability analysis there are three types of eigenmodes with growth rate σ : stable ($\sigma < 0$), unstable ($\sigma > 0$) and neutrally-stable or marginally-stable ($\sigma = 0$). We appreciate much that reviewer #2 further distinguishes neutral eigenmodes into two types: neutral due to symmetry transformations and neutral due to a local bifurcation point. Reviewer #2 is right that we refer to the latter type.

No terminology is established in the community to make a distinction between the two types of neutral eigenmodes, and one needs to put “neutral eigenmode” into an explanatory context. For example “the associated neutral localized eigenmode at the second symmetry-breaking bifurcation” (Lloyd & Sandstede, 2009)[pattern formation community]. Or “using A times the neutral eigenmode at the critical Reynolds number” (Bayly *et al.*, 1988)[shear flow community].

We think we provide sufficient context to understand which type of neutral eigenmode we mean. In the caption of Fig. 3 (former 2) we say “a bifurcation with neutral eigenmode (c) can be detected”. In section “origin of the equilibrium”, we say “On the lower branch of the Nagata equilibrium close to the saddle-node, there is a pitchfork bifurcation at $Re_I = 164$. Its neutrally stable long-wavelength eigenmode is plotted in Fig. 3c.”

In order to remove any possible ambiguity, we added the definition of a “bifurcating eigenmode” to the description in section “origin of the equilibrium” in the revised manuscript:

“Its neutrally stable long-wavelength eigenmode, whose eigenvalue changes sign at Re_I , is plotted in Fig. 3c.”

13. *Remark.*

“suggests the stripe equilibrium to emerge” → “suggests that the stripe equilibrium emerges”

Response. Changed according to the suggestion of reviewer #2.

14. *Remark.*

“periodic at (L_x, L_z) ” → “with periodicity (L_x, L_z) ”

Response. Changed according to the suggestion of reviewer #2.

15. *Remark.*

“subcritical, forward”

These terms are not always used in the same way by all researchers. The authors should explain that they (presumably) mean that

- the new branches (\mathcal{B}) are created for $Re > Re_I$ (forward)
- and that the parent branch (\mathcal{A}) is more stable for $Re > Re_I$ (subcritical), i.e. for $Re > Re_I$, where branch \mathcal{B} exists, it is less stable than branch \mathcal{A} .

But if this is so, why should branch \mathcal{C} (arising from \mathcal{B}) be more observable than \mathcal{A} (the pure Nagata equilibrium, never spontaneously observed in simulations or experiment)? So perhaps the authors mean something else by subcritical?

Response.

We use the terms “subcritical, forward” like reviewer #2 outlines. “Subcritical” means that the bifurcating branch inherits the unstable eigendirection from the parent branch. “Forward” refers to the parameter over which the bifurcation is studied. A reference that illustrates and classifies the possible types of pitchfork bifurcations is Fig. 6 in Tuckerman & Barkley (1990).

We changed “subcritical, forward” to “subcritical, forward in Re ” in the revised manuscript. By doing so, we more explicitly state what we mean with “forward”. A careful reader of this article who could assume that “forward” implies “supercritical” or that “subcritical” implies “backward” will see that the combination of “subcritical, forward in Re ” contradicts these implications. Such a reader must conclude that “subcritical” refers to the stability properties only.

How observable invariant solutions are in a weakly turbulent flow depends on their detailed stability properties. For the stripe equilibrium, these details are discussed in response to remark 1 of reviewer #3.

16. *Remark.*

“The amplitude profiles of single- and double-pulse equilibria show that the single-pulse with period $L_z = 40$ has large modulations . . .

at $Re = 350$ and $\theta = 24^\circ$. . .

Solution branch \mathcal{C} (red line in Fig. 3) reaches $Re_{\mathcal{C}} = 350$ after undergoing an additional saddle-node bifurcation at $(Re, \theta, L_x) = (243, 20.8^\circ, 11.7)$.”

For clarity, I would re-order to:

“Solution branch \mathcal{C} (red line in Fig. 3) reaches $Re_{\mathcal{C}} = 350$ after undergoing an additional saddle-node bifurcation at $(Re, \theta, L_x) = (243, 20.8^\circ, 11.7)$.

The amplitude profiles of single- and double-pulse equilibria

at $Re = 350$ and $\theta = 24^\circ$

show that the single-pulse with period $L_z = 40$ has large modulations . . . ”

Response. Changed according to the suggestion of reviewer #2.

17. *Remark.*

“is not broken by the neutral mode of the primary bifurcation”

Again, rather than “the neutral mode” I would say “by the bifurcating mode” or simply “by the primary bifurcation”.

Response. See response 12 above.

18. *Remark.*

“in spanwise direction” \rightarrow “in the spanwise direction”

Response. Changed according to the suggestion of reviewer #2.

19. *Remark.*

“alignement” \rightarrow “alignment”

Response. Changed according to the suggestion of reviewer #2.

20. *Remark.*

“allows to determine” \rightarrow “determines”

Response. Changed according to the suggestion of reviewer #2.

Response to Reviewer #3

This paper reports the subharmonic instability and the resulting bifurcation on the Nagata (1990) steady solution to the Navier-Stokes equation as well as the subsequent second bifurcation on the found solution branch for plane Couette flow. This second bifurcation leads to a novel large-scale-patterned steady solution to the Navier-Stokes equation, which well describes the spatial property of oblique turbulent bands observed in transitional plane Couette flow. The key contribution of this paper is theoretical identification of the origin of a large-scale oblique pattern appearing in subcritical transition to turbulence in plane Couette flow. It should be highly appreciated because this kind of theoretical identification had not been achieved for commonly observed large-scale turbulence patterns in experiments or numerical simulations. Therefore I could recommend this paper for publication in Nature Communications if the following two comments are taken into account for the revision of the paper:

We thank reviewer #3 for the positive feedback and the two important comments below.

1. *Remark.*

The instability of the novel solution in figure 1 (right) and figure 4 (bottom). Although in page 3 (left), the authors briefly mentioned the instability of this solution, the detailed information should be necessary in the paper. As mentioned there are 15 unstable directions in phase space, but no information has not been provided on the eigenvalues which are indispensable for identification of significance of the instability of the found solution. How large are they with respect to the typical time scale? Without this information we cannot say that it is “weakly unstable”. In addition, the corresponding eigenstructure to the leading (or more) eigenvalue(s) would be mentioned in order to discuss the relevance of the found solution with turbulence dynamics (turbulence dynamics could be considered to be an escaping process out of the invariant coherent solution).

Response.

The manuscript provides the number of unstable (15) and stable ($\sim 10^6$) eigendirections to discuss the stability properties of the stripe equilibrium in the context of a dynamical systems approach to transitional turbulence. We think the fact that the stripe equilibrium attracts the dynamics from almost all directions is sufficient to explain to the general reader the concept of invariant solutions that are linearly unstable but still dynamically relevant.

However, we agree with reviewer #3 that the assessment of the stripe equilibrium as “weakly unstable” must be based on both, the number of unstable directions and their growth rates. Therefore, we added a figure to the revised manuscript that provides the spectrum of eigenvalues of the stripe equilibrium (the new Fig. 2). We feel a detailed discussion of the spatial structure of eigenmodes and the evolution of the nonlinear dynamics when perturbing the equilibrium in various eigendirections is beyond the scope of the current manuscript. The important information is that the solution is indeed not very unstable.

In response to this remark of reviewer #3, we thus explain why the stripe equilibrium is correctly described as “weakly unstable” in the paragraph on the stability properties of the stripe equilibrium. There we say:

“The unstable eigenspace of the evolution operator linearized around the equilibrium

is spanned by 15 directions. The remaining $\sim 10^6$ directions are attracting. Consequently, the dynamics is attracted towards the stripe equilibrium from almost all directions. The exponential growth rates ω_r along the unstable directions are small compared to typical turbulent time scales in the flow, given by the oscillatory frequencies ω_i in the spectrum of eigenvalues (Fig. 2). The low dimensional unstable eigenspace and the small exponential growth rates suggest a weakly unstable exact invariant solution...”

In addition, more information is contained in the caption of the new Figure 2 (here reproduced as Fig. 1) illustrating the small growth rates:

Figure 1: The eigenvalue spectrum characterizes the linear stability of the stripe equilibrium in Fig. 1. Shown are the 100 leading eigenvalues in the complex plane (ω_r, ω_i) , computed by Arnoldi iteration³³. 15 eigenvalues with a positive real part $\omega_r > 0$ quantify the exponential growth rate ω_r along the unstable directions in the linear eigenspace of the stripe equilibrium. The imaginary parts ω_i quantify the oscillatory frequencies in the eigenspace and the absolute values indicate typical turbulent time scales in the flow in units of U/h . The wide aspect ratio of the spectrum (uniform axes) graphically illustrates that the maximum growth rate $\omega_r = 0.034$ is small compared to the typical turbulent time scales. The stripe equilibrium is thus weakly unstable.

2. Remark.

Figure 5 and corresponding statement.

This figure is important to interpret the selection mechanism of the inclination angle of oblique bands. The authors have shown the turning points suggesting the smaller and larger angle ends; however, there might be possibilities that the solution branches would further turn in the decreasing (increasing) direction on the smaller (larger) end. Have the authors confirmed that the solution branches are closed? If so, this should be mentioned. If not, the statement would be weaker than in the present form.

Response.

The range of existence of the stripe equilibrium over the pattern orientation angle θ is limited on both ends by the saddle-node bifurcations shown in Fig. 6 (previously Fig. 5). We have continued the solution branches further beyond the saddle-node bifurcations. Along these branches, the equilibrium solutions change their amplitude modulation from single-pulse to double-pulse profiles and do not resemble the oblique stripe pattern anymore. Thus, the equilibrium solutions beyond the saddle-node bifurcations differ from the stripe equilibrium discussed in the section “Pattern angle

selection”.

We specifically show the additional data at the lowest and the highest Reynolds numbers (see Fig. 2 in this document). At $Re = 200$, the solution branches on both sides quickly undergo symmetry-breaking bifurcations and connect with a branch of a double-pulse equilibrium solution of type \mathcal{B} (cross markers). Together with the \mathcal{B} -branch they form a closed isola. At $Re = 350$, such a connection to double-pulse equilibrium \mathcal{B} is only found beyond the lower angle saddle-node bifurcation. The solution branch beyond the upper angle saddle-node bifurcation is indeed subject to additional folding, as reviewer #3 guessed correctly. These additional folds make the corresponding equilibrium solutions more unstable, reduce the amplitude variations and approach a double-pulse equilibrium without actually connecting to equilibrium \mathcal{B} (see inset in Fig. 2 in this document). The solution no longer resembles a turbulent-laminar stripe.

Therefore, the additional data does not change the statement that we make in the manuscript, namely that the stripe equilibrium only exists in the quoted range of angles. We understand that plotting only the initial part of the bifurcation branches beyond the saddle-node bifurcations (dotted lines in Fig. 6) raises the question of how these equilibrium solutions continue. We therefore include the following description in the caption of Fig. 6 in the revised manuscript:

“Along the remaining parts of the branches (not shown), the equilibrium solutions no longer represent the stripe pattern.”

References

- BAYLY, B J, ORSZAG, S A & HERBERT, T 1988 Instability mechanisms in shear-flow transition. *Annual Review of Fluid Mechanics* **20**, 359–391.
- LLOYD, DAVID J B & SANDSTEDE, BJÖRN 2009 Localized radial solutions of the Swift-Hohenberg equation. *Nonlinearity* **22** (2), 485–524.
- TUCKERMAN, LAURETTE S. & BARKLEY, DWIGHT 1990 Bifurcation analysis of the Eckhaus instability. *Physica D: Nonlinear Phenomena* **46** (1), 57–86.

Figure 2: Parametric continuation of equilibrium solutions along pattern orientation angle θ , like Fig. 6 in the present manuscript (black lines) but including the remaining data of the continued solution branches at $Re = 200$ and $Re = 350$ beyond the first saddle-node bifurcations (red lines). On both sides at $Re = 200$ and on one side at $Re = 350$, the branches terminate in a symmetry-breaking bifurcation from the double-pulse equilibrium of type \mathcal{B} with pattern wavelength $\lambda = 20$ (cross markers). The inset panel shows amplitude profiles at marked points. Obviously, the profiles no longer represent the amplitude variation of the stripe pattern with $\lambda = 40$. Beyond the upper angle saddle-node bifurcation at $Re = 350$, the branch approaches a double-pulse equilibrium but seems to fold indefinitely at low orientation angles.

Reviewer #1 (Remarks to the Author):

The authors have revised the manuscript well and answered the reviewers' questions reasonably. Then I can recommend its publication in Nature Communications.

Reviewer #3 (Remarks to the Author):

The authors have revised the manuscript by taking into account my comments, and thus I could recommend the paper for publication in Nature Communications.

Response 2 to Reviewer #1

The authors have revised the manuscript well and answered the reviewers' questions reasonably. Then I can recommend its publication in Nature Communications.

We thank reviewer #1 for the positive evaluation of our manuscript.

Response 2 to Reviewer #2

My opinion of the article remains quite positive. Here are my remaining minor comments on the revised version.

We thank reviewer #2 for the additional careful comments on the revised manuscript.

1. *Remark.*

“as preferred solution” → “as the preferred solution”

Response. Changed according to the suggestion of reviewer #2.

2. *Remark.*

“allow the simulation of flows but also to construct exact equilibria” → “allow the simulation of flows but also the construction of exact equilibria”

Response. Changed according to the suggestion of reviewer #2.

3. *Remark.*

My previous report highlighted the sentence:

“These dynamically unstable exact invariant solutions lead to a description of turbulence as a chaotic walk among invariant solutions.”

The authors have changed this to:

“Based on these dynamically unstable exact invariant solutions one may describe turbulence as a chaotic walk among invariant solutions”.

This does not address my objection. I stated:

“I wouldnt quite state the theory in this way. The theory from which the chaotic walk is predicted is not a consequence of the existence of many unstable solutions. A system can have many unstable solutions but not execute a chaotic walk between them. In fact, the Navier-Stokes equations always have many unstable solutions in most circumstances above certain Reynolds numbers, even when the solution observed is merely steady or periodic.”

The authors are saying the same thing in both versions: There exist unstable exact invariant solutions and this has some causal relationship with turbulence (“based on” or “lead to” mean the same thing). I disagree. The theory of turbulence as a chaotic walk stands (or does not stand) on its own. The Navier-Stokes equations always have a large number of unstable exact solutions, and it is not the discovery of this or other unstable exact solutions that supports the theory.

Response.

What we want to say here is that the exact solutions play a key role in the emerging dynamical systems view of turbulence. The question raised by reviewer #2 is if we

are giving the wrong impression that the existence of unstable solutions alone or directly implies the existence of turbulence. This would indeed be misleading. To avoid possible misinterpretation and to clearly separate the idea of strange saddles in state space supporting chaotic dynamics from the discovery of invariant solutions that only together yield the picture of a chaotic walk between unstable solutions, we rewrote the paragraph as follows:

“These exact invariant solutions are believed to be embedded in a strange invariant set generating the chaotic dynamics of turbulent flow in the system’s state space [Ref. 24]. Consequently, a picture emerges where turbulent flow is described as a chaotic walk between dynamically unstable invariant solutions which together with their entangled stable and unstable manifolds support the turbulent dynamics [Ref. 25].”

4. *Remark.*

I find interesting the clarification/quantification provided by the authors of the meaning of weak instability for this new solution, i.e. small positive real parts compared to larger imaginary parts for the eigenvalues.

My previous report asked: “why should branch \mathcal{C} (arising from \mathcal{B}) be more observable than \mathcal{A} (the pure Nagata equilibrium, never spontaneously observed in simulations or experiment)?”

Although the authors did not answer this question, I think I now understand that the saddle-node bifurcation is responsible for this stabilization. That is, branch \mathcal{B} created at I has the same stability as \mathcal{A} , the Nagata solution with, say, N unstable eigenvalues. When the backwards branching \mathcal{C} is created at II, it initially has $N - 1$ unstable eigenvalues (as does branch \mathcal{B} to the right of II). The saddle node at $\text{Re} = 243$ changes the number of unstable eigenvalues from $N - 1$ to either N or to $N - 2$. So branch \mathcal{C} has either the same stability (N) as the Nagata solution or has two fewer ($N - 2$) unstable eigenvalues. Is this true? If so, which one is the case? Since the authors mention the stability of the new solution, all of this might be worth mentioning briefly, i.e. that the new states are precisely as stable as (or have two fewer unstable eigenvalues than) the Nagata solution.

Response.

Reviewer #2 wants to know how exactly the numbers of unstable eigendirections change across the bifurcation diagram. The two pitchfork bifurcations (I and II) and the saddle-node bifurcation at $\text{Re} = 243$ change the number of unstable eigendirections N in the following ways.

Pitchfork I: Below threshold, \mathcal{A} has $N = 3$. Above threshold, \mathcal{A} has $N = 2$, and \mathcal{B} has $N = 3$. Since the bifurcating branch has one unstable eigendirection more than the coexisting parent branch, the bifurcation is subcritical as stated in the text.

Pitchfork II: Below threshold, \mathcal{B} has $N = 13$ and \mathcal{C} has $N = 14$. Above threshold, \mathcal{B} has $N = 14$. Since the bifurcating branch has one unstable eigendirection more than the coexisting parent branch, the bifurcation is subcritical as stated in the text.

Saddle-node: The ‘upper branch’ of \mathcal{C} emerges with $N = 8$. The ‘lower branch’ of \mathcal{C} emerges with $N = 7$.

All numbers hold only very close to the bifurcation points because the number of unstable eigendirections further changes along the solution branches, indicating the presence of additional bifurcations. The reviewer’s counting of eigenvalues across the bifurcation diagram implicitly assumes the absence of additional bifurcations. In contrast to low dimensional dynamical systems, these additional bifurcations are a common fea-

ture of exact solutions in high dimensional problems like 3D fluid flow. Consequently the number of unstable dimensions needs to be analyzed at each bifurcation point individually.

A complete discussion of how the number of unstable eigendirections varies across the bifurcation diagram is thus more involved than reviewer #2 seems to suggest. We feel such a discussion would be too long to be included in the very focused manuscript. Moreover, the number of unstable directions provides only an incomplete characterization of the stability features. In addition one would need to discuss the changing magnitude of the eigenvalues (see response to reviewer #3). Such a detailed and technical discussion of spectra along the solution branches in this letter would dilute the message of the manuscript.

5. *Remark.*

Figure 4 caption: Since elsewhere, parameter-space coordinates are given as (Re, θ, L_x) , I recommend that the same be done here, instead of just $(\text{Re}, \theta)_I$ and $(\text{Re}, \theta)_I$, just for consistency with the text.

Response. Changed according to the suggestion of reviewer #2.

6. *Remark.*

Figure 5 caption: Shouldnt the streamwise vorticity be written as the dot product $\omega' = (\cos(\theta)\mathbf{e}_x + \sin(\theta)\mathbf{e}_z) \cdot \nabla \times \mathbf{u}$ rather than as $\omega' = \nabla \times \mathbf{u} (\cos(\theta)\mathbf{e}_x + \sin(\theta)\mathbf{e}_z)$?

Response. We agree. Changed according to the suggestion of reviewer #2.

7. *Remark.*

In figure 5, it looks as though the streamwise direction (arrow on top of diagram \mathcal{A}) is parallel to the red arrow shown in diagram \mathcal{C} and labelled as θ . Shouldnt an angle be between two lines and isnt θ the angle between the streamwise and the x direction?

Response.

The arrows in panel \mathcal{C} mark the angles between the wave fronts (thick red/blue lines) and the spanwise direction (grey grid). In the figure caption we say:

“The stripe equilibrium \mathcal{C} has sigmoidal wave fronts. In the turbulent region the wave fronts are oriented at $\theta = 24^\circ$ (red/blue arrows), and align with the pattern wave vector (in the z -direction).”

The angle between the wave fronts in the turbulent region and the spanwise direction is identical to the angle θ between the streamwise and the x direction. Thus, the description is correct.

8. *Remark.*

the wave fronts of the wavy streaks are skewed at $\theta = 24^\circ$ against the spanwise direction. Is this phrase clear? (Maybe it is.)

Response.

The orientation of the wave fronts, that we define as “lines connecting vorticity maxima or minima in the spanwise direction”, changes along the bifurcation branches. We consistently describe the orientation of the wave fronts relative to the spanwise direction:

“Straight and strictly spanwise oriented wave fronts indicate a constant streamwise phase of all streaks of the Nagata equilibrium \mathcal{A} ”.

“In the turbulent region of equilibrium \mathcal{C} , the wave fronts of the wavy streaks are

skewed at $\theta = 24^\circ$ against the spanwise direction and oriented exactly along the pattern wave vector (C in Fig. 5).”

Consequently, the sentence to which reviewer #2 refers is unambiguous.

9. *Remark.*

“This suggests a selection mechanism for the pattern angle that explains why turbulent-laminar stripes are oblique.”

Certainly, this research adds a great deal to our knowledge of these states, but I am not sure I would agree that the authors have found a “selection mechanism” or know “why” the stripes are oblique.

Response.

The existence of the stripe equilibrium is a major step towards understanding the structure of self-organized oblique stripe patterns. Of course open questions remain, including identifying other exact invariant solutions underlying the chaotic dynamics within the stripes. As stated at the end of section “origin of the equilibrium”, we hypothesize that additional invariant solutions with similar streaky structure like the Nagata solution follow the same angle constraints as the stripe equilibrium. Assuming that the angle selection mechanism found here can be transferred to other relevant invariant solutions underlying the dynamics, our work suggests a mechanism that selects the angle of the chaotic turbulent-laminar pattern. In order to avoid giving the expression of over-stating our results, we modified the statement slightly to imply that not all questions have been answered but that future research needs to confirm the hypothesis, that the angle selection within the shown equilibrium is characteristic of other dynamically relevant invariant solutions:

“This suggests a selection mechanism for the pattern angle and provides a route towards explaining why turbulent-laminar stripes are oblique.”

Response 2 to Reviewer #3

The authors have revised the manuscript by taking into account my comments, and thus I could recommend the paper for publication in Nature Communications.

We thank reviewer #3 for the positive evaluation of our manuscript.